# Improving Energy Access in Low-Income Sub-Saharan African Countries: A Case Study of Malawi

**Ehiaze Augustine Ehimen** [1,*], **Peter Yamikani Sandula** [2], **Thomas Robin** [1] **and Gregory Tsonga Gamula** [2]

1   Centre for Environmental Research Innovation and Sustainability (CERIS), Atlantic Technological University, F91 YW50 Sligo, Ireland
2   Department of Electrical Engineering, Faculty of Engineering, Malawi University of Business and Applied Sciences, Blantyre 312225, Malawi
*   Correspondence: ehiaze.ehimen@atu.ie; Tel.: +353-894524313

**Abstract:** The inaccessibility of modern energy in low-income countries (LIC), especially in the Sub-Saharan African (SSA) region, continues to be a problem in the 21st century. The lack of access to modern energy has led to an inability to implement developmental structures and initiatives. While considerable progress and successes have been realised in the last three decades with increased activities and global commitments from international governments and multinational agencies through electrification projects in the SSA region, SSA countries remain off-track in their efforts to achieve the UN Sustainable Development Goal (SDG) 7. This is mainly demonstrated in sparsely populated rural regions where the high cost of centralised power generation, poor transmission and distribution infrastructures and economic factors have been a major barrier to electricity expansion. Although the use of RES (i.e., decentralised or stand-alone systems) have been acknowledged by the International Energy Agency (IEA) to be the least expensive route to improving access, its impact has not been effectively demonstrated regionally. Decentralised RES use in SSA countries have not seen significant uptake and/or enjoyed long-term sustainability owing to a number of factors. Malawi, despite its significant hydropower resources and the favourable proximity of its inhabitants to grid infrastructures, still has one of the lowest levels of access to electricity globally, with 86% of the population having no access to electricity. The country provides a good case study to investigate factors limiting electricity access in SSA countries. This paper explores the main issues that have historically hindered the uptake and sustainable operation of decentralised RES in the country. Recommendations to facilitate a potential improvement in RES use as a pathway to improved universal energy access are then put forward.

**Keywords:** energy access; renewable energy systems; Sub-Saharan Africa (SSA); Malawi; low-income countries; power generation; electricity

## 1. Introduction

The inaccessibility of modern energy (electricity) to meet the basic needs of lighting, food processing, education, health centres and income generation industry demands of populations in low-income countries (LICs) continues to be a problem in the 21st century. While governments of industrialised countries appear to be focused on global oil and natural gas prices, concerns over energy security, climate change and the need for more alternative energy contributions to total energy supply, this energy crisis (inaccessibility and significant energy poverty), affecting millions of people in LICs, has been largely ignored. The lack of access to electricity has led to the inability of government and institutions to implement developmental structures and initiatives, which has in turn condemned millions of men, women and children to continue living in absolute economic poverty. In 2015, the United Nations (UN) General Assembly outlined 17 Sustainable Development Goals (SDG), intending to improve the quality of life of populations across the planet. Among them, SDG

7 targets the delivery of affordable clean energy for all by 2030. A total of 1.2 billion people were reported to have no direct access to electricity as of 2010, with this number decreasing to 733 million in 2020, which falls short of the initial target defined by the UN [1]. In 2020, it was estimated that over 600 million inhabitants in the region have no direct access to electricity, representing 77% of the global population without electricity access [2]. The situation is particularly critical in remote and rural areas. This means no electrical lighting in homes, limited access to radio and modern communications, inadequate utilisation of modern and digital education support systems, poor health facilities and services (i.e., with limited refrigeration capacity for vaccines and inability to carry out medical procedures) and insufficient power to support businesses for such communities. While considerable progress and successes have been realised, especially in the past two decades, through electrification projects which have led to a greater accessibility to electricity than ever before, the world still remains off-track in its efforts to achieve the UN SDG 7 objectives. The SSA region appears to be where most work is still required, with 15 countries in the region having access rates below 25% [3]. Furthermore, recent events, such as the COVID pandemic, coupled with the energy crisis resulting from the Russia–Ukraine conflict has brought about a slowdown in efforts and programmes aimed at achieving improved energy accessibility for SSA LIC communities, including Malawi [4]. A reduction of the annual investment globally for electrification projects has been observed, dropping from USD 24.7 billion (in 2017) to 10.9 billion (in 2019) [5], with forecasts indicating that such investments will see a reduction after 2020. The SSA region is reported to require a total investment of USD 45 billion to USD 49 billion by 2030 to ensure universal access to a source of electricity [6]. Although current investment levels are off target, there are renewed expectations of improved capital flows to facilitate the target realisation, especially with the mechanisms put in place through the African Development Bank's New Deal on Energy for Africa (NDEA) and Sustainable Energy Fund for Africa (SEFA) [7]. International agencies and government activities, as well as campaigns and projects funded and executed in the SSA regions, have further helped increase the expansion of electricity access and implementation of renewable energy solutions.

Historically, mechanisms to facilitate improvement in electricity access to populations have been mainly implemented through the expansion of existing grid systems and an increase of the large-scale power generation capacities of such countries. However, the unfavourable costs, operational limitations and issues such as pilferage and limitations related to the use of centralised power management systems, especially for cases where there are small load requirements, could render the use and expansion of existing grid infrastructures unattractive for remote places, and in some cases impractical [8]. With SSA rural areas, typically with lower load densities, experiencing inaccessibility problems more frequently, this proves problematic since rural electrification has been reported to be typically more expensive (than urban areas) due to lower capacity utilisation rates, greater electricity transmission costs and higher infrastructure and maintenance costs [9]. The cost of electricity being delivered to a given location directly relates to the load factor, transmission and distribution losses and the electricity generation cost; therefore, increased distribution lines, low load densities and high transmission losses would make the expansion of conventional grid systems to remote rural areas economically unsuitable [8].

Alternative systems or a combination of mechanisms to afford the ability to access an electricity source in such currently underserved communities are therefore required, although grid extensions are still being pursued in SSA countries, with large-scale power generation infrastructures and the distribution backbone already in place, so its extension is a more ideal option where practical. Off-grid solutions, including the installation and operation of renewable energy systems (RES) such as solar PV and wind generation systems, are increasingly being used to provide electricity access. Off-grid systems (which can function independently without a connection to the central grid network) can be categorised into distributed and decentralised systems. The distributed off-grid systems are usually characterised by having a power distribution network, similar to a mini-grid, and the

decentralised systems mainly find applications in a particular location (home or community use) and encompass solutions such as solar or micro-wind systems for single home use and community grid systems [10]. The installation and operation of community-based decentralised off-grid RES has been reported to be critical for the achievement of universal electricity access; this is especially significant in sparsely populated rural regions where the high cost of centralised power generation and transmission infrastructures has been a huge barrier to electricity expansion to such areas. Furthermore, the use of such decentralised RES have been acknowledged by the IEA to be the least expensive route to ensuring power provision and improving access to more than half of the currently deprived populations by the year 2030. Even with such optimistic projections, the installation of decentralised RES systems, especially in SSA countries, has still not experienced significant uptake owing to a number of factors, resulting in the possible non-achievement of the SDG 7 targets by 2030. With the reduced prices of solar PV and battery systems in recent years, the investment of micro-grids has increased; in 2016, the RE market was worth over USD 200 billion each year [11,12]. However, the implementation of RES to improve electricity access in SSA LICs is plagued by low private sector engagement notwithstanding the tremendous future growth potentials, with investors deterred by the lack of profit in the short–medium term. This is mainly due to the low-income rates of the inhabitants of such regions and the inability of such communities to economically sustain such projects. With only about 14% of the national population having access to electricity (2020 estimates), Malawi has one of the lowest electricity access rates in the world. With a population of approximately 18.1 million (2018 figures), 83.1% of the inhabitants of Malawi live in rural areas (with a density of 192 people per km$^2$) [13]. The population growth rate for the country has been observed to be increasing significantly over the last few years, with the population expected to reach 25 million by 2025 (with a growth rate of 2.8%) ([13,14]). The Malawian economy is highly dependent on the agriculture industry, with over 80% of the population working in this sector ([13]). The agriculture sector contributes roughly 34% of the gross domestic product of Malawi [15]. With the majority of the activities in this sector taking place in rural settings, electricity inaccessibility is expected to negatively impact the processing and preservation capabilities related to agricultural production which would in turn have improved the contributions of agriculture to the economy. Malawi is currently reported to have one of the lowest GDPs in the world (the third lowest) with approximately USD 349 per capita (2018), which is below the regional average (USD 4098 (ppp)) [16]. The rate of poverty is high and observed to exceed 50%, especially in rural areas, where the vast proportion of the population are currently living on less than USD 2/day [17].

The prevailing widespread non-accessibility of electricity and the economic characteristics of Malawi, therefore, make it a suitable candidate to explore the mechanisms (current and proposed) put in place to facilitate electricity expansion. Evidence-based research conducted across varied communities, i.e., from Zimbabwe [18], Brazil [19] and Indonesia [20], has shown the close relationship between electricity access and poverty alleviation. The current state of play regarding electricity provision (including the energy generation and supply issues), national electrification programmes, challenges, and situational analysis to support the universal electrification goals of Malawi will be covered in this paper. While there has been some research output on the issue of electricity inaccessibility in Malawi and the electricity landscape in the country, e.g., [21], a lack of academic research papers covering situational assessments of specific national mechanisms, and their limitations to affording improvements in electricity access in Malawi, was observed.

This paper intends to address this gap identified in the literature. The paper will provide an understanding of the particular situation regarding the overall power generation and distribution situation in Malawi. Furthermore, the challenges for the persistence of low electrification rates nationally will be investigated, with a specific focus on RE initiatives that have been pursued, and a reflection on why electricity expansion using decentralised RES has not been successful historically. The policies and government mechanisms supporting energy generation and RE implementation will be covered in the paper and used as a basis

to evaluate the systemic issues impacting the uptake and use of such technologies in Malawi. Recommendations and discussions on how improvements to national electrification can be attained using decentralised RES are then put forward. Although the analysis is initially focused on Malawi, it is anticipated that the findings and assessments put forward in this paper will be similar to the experiences of and be used as a pointer for potential accessibility for, other SSA LIC communities.

## 2. Overview of Electricity Generation and Demand in Malawi

This section presents an overview of the electricity sector in Malawi. A reflection on the electricity landscape (including supply and national capacities) of the country, the electricity market structure and policies related to electrification are covered.

### 2.1. Malawi's Electricity Supply and Grid System

National electricity supply in Malawi dates back to the 1950s when electricity was generated from a coal-powered plant at what is now called the ESCOM Power House in Blantyre and a mini hydropower station at Mulunguzi River in Zomba (the former capital of Malawi) [22]. The generation capacity was increased when the Nkula A Power Station was built between 1966 and 1967 on the Shire River in Neno District. Between 1966 and 2021, additional power plants were constructed with around 485.7 MW capacity added to the national electricity grid, as indicated in Table 1. Figure 1 highlights the current locations of the main power generation facilities in Malawi.

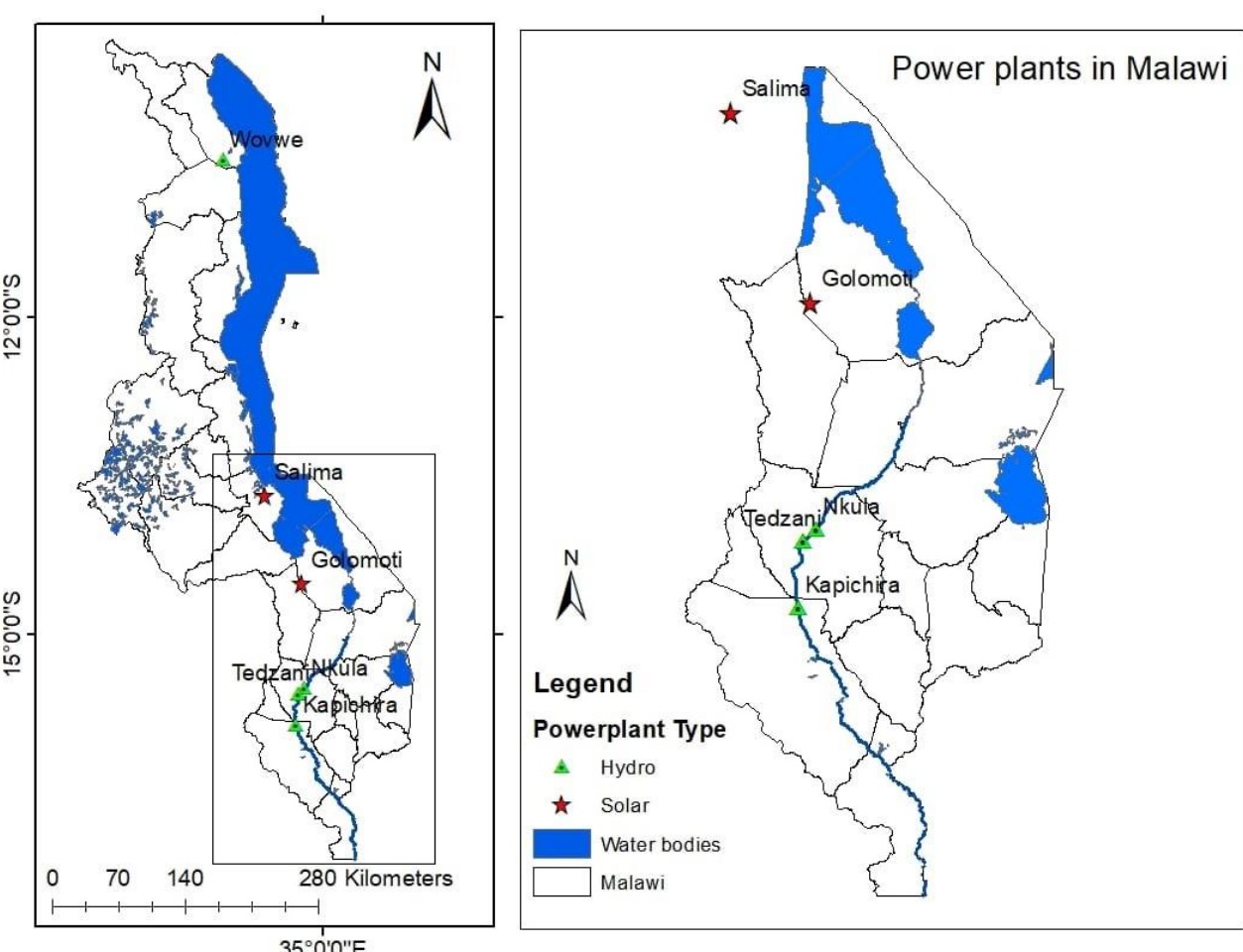

**Figure 1.** A map indicating the locations of power generation facilities in Malawi.

**Table 1.** Malawi's total installed capacity (adapted from [23]).

| Power Station | Installed Capacity (MW) | Year Commissioned | Annual Inflation Rates (%) [24] |
|---|---|---|---|
| Nkula A and B Hydropower Station | 135.10 | 1966–1992 | 0.2–13.3 |
| Kapichira Hydropower Station | 129.6 | 2000–2014 | 30.5–20.9 |
| Tedzani Hydropower Station | 102.0 | 1973–1996 | 9.3–52.3 |
| Thermal Power Plants | 53.2 | 2019 | 7.7 |
| Wovwe Hydropower Station | 4.5 | 1996 | 52.3 |
| Likoma and Chizumula Solar Power Plant | 1.3 | 2020 | 10.2 |
| Salima Solar PV power plant | 60 | 2021 | 8.5 |
| Total Installed Capacity (2021) | **485.7 MW** | | |

As noted, 75% of the total installed capacity is from hydropower resources, with most of the remainder previously produced by the state-owned electricity generation company EGENCO, and thermal power plants (diesel power generators) are usually used for peak load. All major hydropower stations are in the southern region of the country along the Shire River. There is, however, a small hydro station, the 4.5 MW Wovwe plant, which operates in the Karonga district in the northern part of Malawi [25]. According to [26], Malawi's annual population growth rate is 2.8%, with an urbanisation rate of 4.2%, which consequently indicates that future demand for electricity is projected to outstrip supply. The supply and demand trends observed in Malawi's power sector indicates a significant power shortage in the foreseeable future due to an increase in electricity demand averaging 7% per year [18]. Recently, as part of efforts to diversify the electricity sources in the country and increase the generation capacity, there has been an increase in the electricity generation using solar PV, with the installation and operation of the 60 MW Salima Solar PV power plant (near Lilongwe). As of 2021, the country's total installed capacity was 485.7 MW, with an available capacity of 263.30 MW, which falls short of the current national connected demand of 700 MW [23]. The significant difference in installed and available capacity is attributed to power loss at the Kapichira hydropower plant, which was damaged by a cyclone [23]. The electricity sector has been further incapacitated by the aging transmission infrastructure, which has also resulted in power losses such that there is a significant difference between installed capacity and available capacity. To address this challenge, transmission networks have been upgraded such that the transmission system presently comprises some 1340 km of 132 kV lines and 1100 km of 66 kV lines and associated substations. Total system losses have seen a major improvement from 21% in 2012 to 13–18% between 2016 and 2017 [11]. This available generation capacity does not meet the national electricity demand, as only 14% of the population have access to electricity, of which 11.4% is from the national electricity grid and the rest mainly from solar power systems and batteries [27]. The situation is particularly worrisome in the rural areas of Malawi, which have an access rate of 3.9% [28].

With a significant dependence on hydropower to meet the current supply levels, efforts to increase electricity generation to meet demand have been met with challenges such as the high wear and tear of electro-mechanical parts on the current hydro-generation plants due to the sandblasting effect following environmental degradation along the waterways. Other challenges include inadequate plant maintenance, sedimentation, an infestation of aquatic weeds in the reservoirs, frequent storms and floods resulting in dam flooding and siltation of the river system causing low storage capacity [22]. As a way of addressing the challenge of inadequate generation capacity, in 2016, the Government of Malawi (GoM) unbundled the Electricity Supply Corporation of Malawi, ESCOM, to form a new state-owned Electricity Generation Company, EGENCO, which took over the generation assets from ESCOM [29]. The new market structure also allows Independent Power Producers, IPPs, to generate electricity and feed into the national grid. The new market structure, coupled with the opening up of the power generation sector, has led to significant progress being made in the last decade in relation to electrification expansion in Malawi. This has

particularly been driven by international collaboration projects. For example, between 2015 and 2018, the Malawian Ministry of Energy (MoE) rolled out a three-year project funded by the Government of Scotland called the Sustainable Off-Grid Electrification of Rural Villages Project (SOGERV), which explored the potential to deploy solar PV off-grid solutions in the Chikwawa District (Gola, Kande, Mandrade and Thendo) of the country [13]. Additionally, the United Nations Development Programme (UNDP) has been involved in various projects aimed at increasing access to clean and affordable energy services via mini-grids in selected vulnerable areas of Malawi. With support from the UNDP, the MoE revised its energy policies and legislative and regulatory frameworks to create a conducive space for private sector participation in electricity generation and supply through grid-connected and off-grid renewable energy systems, such as mini-grids and solar home systems [30].

Table 2 presents a summary of a list of projects nationally funded by various developmental partners with a goal to enhance electricity access through off-grid solutions.

In the longer term, Malawi has the potential for expanding its generation capacity through increasing investment in its power generation and transmission infrastructures, as well as through investing in electricity importation from neighbouring countries using interconnectors. The Malawi government, through the MoE, has engaged with various development partners for potential investments in its energy generation and supply sector, with the initial focus on hydropower plants for increasing the national generation capacity. This was due to the significant hydro resources in the country. Table 3 presents the ongoing and earmarked power generation projects.

**Table 2.** List of off-grid and grid-connected renewable electricity projects aiming to support the acceleration of energy access in Malawi (adapted from [13]).

| Donor | Funding | Dates | Technology | Objectives |
|---|---|---|---|---|
| **UNDP** | USD 4 million | 2015–2021 | Mini-grid | Aimed at increasing access to electricity in selected rural areas in Malawi through the promotion of innovative, community-based mini-grid applications in collaboration with the private sector. |
| **DFID-African Enterprise challenge fund** | USD 2 million | Up to 2018 | Solar PV (mainly Pico) | Concessional loans and grants. It was noted that the project was not successful in promoting the adoption of Solar Home Systems (SHS) due to inadequate funding. |
| **USAID through its Power Africa Programme Kickstarter** | USD 1.5 million | 2019–2021 | SHS, Improved Cooking Stoves (ICS) | The objective was to catalyse and stimulate the market in the short term over 3 years (2019–2021). In terms of the result-based finance (RBF) mechanisms, the institution only disbursed the grant once the energy service company made sales—the sponsored companies would receive the subsidy from the USAID grant facility after financing and selling the systems. |

**Table 2.** *Cont.*

| Donor | Funding | Dates | Technology | Objectives |
|---|---|---|---|---|
| **World Bank Malawi Electricity Access Project—Off-Grid Market Development Fund** | USD 30 million | 2019–2023 | Mini-grids, SHS, ICS | The Off-Grid Market Development Fund by the World Bank is the second component of the World Bank's Malawi Electricity Access Project and is a follow-on to the USAID programme. It was designed in close cooperation with USAID and has three financing components: a mini-grid facility that will be delivered to mini-grid enterprises on completion of project preparation, revolving working capital loans and an RBF grant facility that will deliver loans and grant financing to off-grid solar enterprises. |
| **World Bank Malawi Electricity Access Project—grid extension programme** | USD 105 million | | Grid densification | This component will provide credit for cost-effective, priority investments in grid electrification by supplying households living near existing distribution infrastructure and leveraging geospatial analysis to maximize the number of connections under the financing. Finally, the third component is the technical assistance and implementation support to EDM and FUNAE. The project aimed to connect 140,000 households to the national grid and an additional 100,000 households to off-grid systems. |

**Table 3.** Ongoing or earmarked capital investments in power generation projects (adapted from [30]).

| Project | Cost | Funders | Installed Capacity | Comments |
|---|---|---|---|---|
| **Tedzani IV hydropower plant** | USD 56.1 million | Japanese International Cooperation Agency (JICA) and EGENCO | 18.5 MW | The plant is located in the southern part of Malawi, along the Shire. The project started in July 2017 and was completed in September 2021. |
| **Fufu Hydropower plant** | USD 702 million | Yet to be identified | 261 MW | Located in the Northern Region of Malawi on the South Rukuru river. The construction period will span 5 years commencing in 2024 and is to be fast-tracked depending on the availability of financing. |

**Table 3.** *Cont.*

| Project | Cost | Funders | Installed Capacity | Comments |
|---|---|---|---|---|
| **Mpatamanga Hydropower plant** | USD 1 billion | IFC, SCATEC (A Norwegian-based RE company) | 350 MW | The government has 30% shares in the project, while International Finance Corporation (IFC) holds a 15% stake, with the other 55% shareholding being under Scatec ASA's Hydropower Joint Venture and EDF. Feasibility studies for the five-year project started in 2021. |
| **Kholombidzo** | - | - | 210 MW | The African Development Bank supported the Government of Malawi through MoE to conduct the Feasibility Study of the Kholombidzo hydropower electric plant to determine the potential power to be generated and the feasibility of the development of the power plant. |
| **Malawi–Mozambique interconnection project** | USD 88 million | World Bank-International Development Association (IDA) | 50 MW | The feasibility studies for the Malawi–Mozambique interconnector have been completed. The scope of the Project is to interconnect the Mozambique and Malawi Power Systems at 400 kilovolts(kV) through a 76 km transmission line to be constructed from Matambo Substation in Tete, Mozambique to Phombeya Substation in Balaka District, Malawi. Malawi aims to initially import 50 MW. |

The planned projects can be observed to mainly be targeted at grid capacity expansion, which is practical, since the majority of Malawians (over 90%) live less than 10 km from an existing national grid line. However, the use of off-grid RES systems still has an important role to play not only in remote, difficult-to-electrify areas (i.e., the 3.7 million people living in communities >10 km from the grid, as recommended for implementing decentralised off-grid RES in [13]), but also as a short–medium-term solution for communities which are in grid coverage areas but still are yet to gain access. An overview of national mechanisms and policies in place to facilitate and support RES uptake in Malawi is provided in the following section.

*2.2. Malawi Electricity Market Structure and Policy Guidelines Relating to Decentralised RES*
2.2.1. Malawi Electricity Market Structure

The overall mandate of MoE of Malawi is to ensure the sustainable development and utilisation of energy resources for the socio-economic growth and development of the country. The Malawi Energy Regulatory Authority, MERA, was established by the Energy Regulatory Act (2004) with the mandate of regulating activities in the energy sector fairly and transparently for the benefit of consumers and operators. Key roles performed by MERA include (a) reviewing tariff applications from the Electricity Supply Corporation of Malawi (ESCOM) and recommending tariff changes to the GoM and (b) granting licenses for generation and distribution operators under the 2004 energy legislation. A result of

the Electricity Act (2016), which replaced the Electricity Act (2004), led to the un-bundling of ESCOM, which led to the creation of two companies; namely, Electricity Generation Company Limited (EGENCO) and Power Market Limited, PML. However, PML, which was recently established to play the role of a single buyer, was dissolved in December 2022. There are therefore now two operational energy companies in Malawi owned by the state: EGENCO, which is responsible for electricity generation, and ESCOM, which is responsible for power distribution via the national grid. The single-buyer functions are still performed by ESCOM, and EGENCO is responsible for power generation alongside independent power producers, IPPs, which are privately-owned companies that generate power and feed into the national grid or supply to specific communities. The new electricity market structure has so far added 80 MW to the national electricity grid. This additional power has mainly been generated from solar power plants that were owned and operated by the IPPs. IPPs that do not feed generated power to the grid supply power to specific communities and sell electricity directly to the households in those particular communities. MERA is the energy regulatory body and therefore provides operating licenses to the IPPs. The relationship between ESCOM and IPPs is that ESCOM buys electricity from IPPs that feed electricity into the national grid. SAPP is the Southern African Power Pool. This allows countries in southern Africa to export/import electricity (interconnection) to/from other countries. For instance, Malawi imports 20 MW from Zambia and intends to import power from Mozambique in the near future. Figure 2 represents the electricity market structure in Malawi.

**Figure 2.** Electricity market structure in Malawi (adapted from [25]).

### 2.2.2. National Energy Policy 2018

The National Energy Policy, NEP, was launched in 2018 as a successor to the 2003 national energy policy with the goal of **"increasing access to affordable, reliable, sustainable, efficient and modern energy for every person in the country"**. The policy categorised energy sources as: (1) Electricity from Non-Renewable Sources; (2) Electricity from Renewable Sources; (3) Biomass; (4) Petroleum Fuels; (4) Biofuels; (5) Liquefied Petroleum Gas (LPG); (6) Biogas and Natural Gas (NG); (7) Coal; and Electricity from Nuclear Energy [1]. Priority areas related to renewable energy systems are discussed in priority areas 1 and 8. Priority area 1 (electricity) plans to intensify the electrification of rural trading centres, as well as villages, by providing funding from the Rural Electrification Fund to off-grid rural electrification schemes. Finally, the NEP states that under priority area 1, the government will promote the use of renewable energy technologies and the manufacture of renewable energy products, such as solar panels. Although the priority 1 targets of the NEP have been laudable, there have not been any significant structures put in place nationally to facilitate the realisation of home-grown manufacture of RES products and technologies

such as solar modules. Almost all the systems used in the country are obtained by imports. Specifically, here, priority area 1.4 highlights objectives aiming at rural electrification and puts forward initiatives to support the restructuring of rural electrification management (by the establishment of a rural electrification agency), financing of transformers and infrastructures to aid rural electrification and the funding of off-grid solutions to expand reach (also administered by the rural electrification fund). The implementation and use of RES are covered in priority area 1.5 of the NEP, with the strategies to be employed by the GoM to facilitate the improved uptake of such technologies, promoting community-scale RE initiatives and power generation, capacity building in the sector and improving public and private sector partnerships covered in the policy document. Priority area 8 (demand-side management) outlines the intention to institute appliance testing, labelling and standards, which will include minimum energy performance standards; reducing or eliminating import duty and taxes on energy-efficient products; conducting public information campaigns to raise awareness amongst consumers; the provision of financing for energy efficiency measures, allowing consumers to repay loans as part of their utility bills; and promoting energy-saving electrical and biomass-fuelled devices. While the NEP document is quite extensive and laudable in the objectives and the strategies put forward to achieve the goals related to rural electrification and specifically off-grid RES use being adequately outlined, most of the identified policy strategies remain unrealised. For example, strategies such as the provision of net metering to all stand-alone RE-powered mini-grids and private installations, putting in place incentives to increase the number of RE-related workforce and financing schemes for the private sector to manufacture RE products, which would have significantly positively impacted the uptake of RES off-grid solutions and in turn aided improved electrification, have not yet been implemented, and no structures have yet been devised to operationalise these aspects of the policy. The NEP therefore only provides a starting basis for the formulation of several relevant strategies, framework and policy initiatives aimed at consolidating and actualising the energy of the national plan, with the aim of achieving universal access to cleaner energy for all. Such strategies and frameworks relevant to the realisation of, and improving, off-grid decentralised RES uptake and operation are presented in the subsequent sections.

2.2.3. Malawi Renewable Energy Strategy

The Malawi Renewable Energy Strategy (MRES) follows on from the NEP and aims to deliver on the renewable-energy-focused aspects of the policy document. In addition to enhancing the capacity of grid-scale renewables in the country, the objective of this strategy is to ensure universal access to renewable electricity and a sustainable bioenergy sector [31]. The MRES specifies the actions required to deliver the above objective, focusing on clean electricity mini-grids, off-grid power, less polluting cookstoves, solid biofuels, biogas and biofuels in transport. With the MRES, the short-term actions for RE mini-grids have been identified, but vital mechanisms which are necessary for the efficient uptake and market capitalisation especially regarding privately owned and operated RE decentralised mini-grids were observed not to have commenced and, in most cases, did not have any funding allocated to its realisation in the strategy. This includes the need for a viable feed in tariff for RE-generated power in Malawi. Furthermore, actions such as the setting of standards to ensure quality regulation, and the clarification of mini-grid standards and licensing requirements, are also conspicuously absent from the strategy document. This therefore provides significant uncertainty to potential investors and community users since unfavourable regulations might be put in place after implementation. Regarding the bioenergy, biomass and biogas aspects of the strategy, emphasis was placed on the use of these resources for cooking fuel production and transportation requirements. The potential of power generation using micro-biomass-fired generators or biogas power plants were not considered. With the vast biomass and waste resources in the country, the integration of this renewable and sustainable resource in future electricity generation plans could have been included.

### 2.2.4. Mini-Grids Regulatory Framework

Published in July 2020, this framework document furthers the regulatory, licensing and oversight intentions of the NEP regarding mini-grids, and addresses some of the issues raised regarding licensing and related issues for mini-grids in the MRES as discussed in Section 2.2.3. The regulatory framework for mini-grids is intended to achieve the sustainable development and operation of mini-grids in Malawi while striving towards the provision of modern energy services to remote communities where grid extension is not economically feasible [32]. Building from the commitments of the National Energy Policy and the Renewable Energy Strategy, the regulatory framework sets out the guidelines for the development and operation of mini-grids in Malawi covering the following key areas: design considerations; solicitation process; requirements for approval; terms and conditions for licensing and registration; governance structures; quality of supply and service standards; tariff methodologies and structures; and linkages with the national grid. The regulatory framework therefore provides a valuable guide for the requirements necessary for the development and operation of mini-grids in Malawi. Critically, concessions and waivers have been included for sites with generation capacity under 150 kW (i.e., no license or registration required), and provision is made for a variety of subsidies that either reduce consumer tariffs, lower capital expenditure, or incentivise new connections, which are in turn expected to increase rural electrification rates. Private mini-grids over 150 kW, however, shall be registered for record purposes. Table 4 summarises the annual licensing and registration fees based on mini-grid system capacities and purposes.

**Table 4.** Mini-grid license and registration fees in Malawi (adapted from [32]). MK = Malawian Kwacha.

| System Capacity | Purpose | License Application Fees (MK) | Registration Fees (MK) | License Fees (MK) |
|---|---|---|---|---|
| <150 kW | Private | None | None | None |
| <150 kW | Commercial | None | 50,000 | None |
| =150 kW–<1 MW | Private | None | 100,000 | None |
| =150 kW–<1 MW | Commercial | 40,000 | None | 250,000 |
| 1–5 MW | Private | None | 150,000 | None |
| 1–5 MW | Commercial | 50,000 | None | 500,000 |

The establishment of a regulatory framework sets a solid foundation for scaling up mini-grids in Malawi and providing developers with the clarity and reassurance to invest in new projects.

### 2.2.5. Malawi National Electrification Strategy

The National Electrification Strategy (NES) proposes a framework through which the GoM will guide accelerated access to households and businesses at acceptable quality and levels of service that are anchored in the priority policies presented in the NEP 2018 [2]. The strategic elements are summarised in Table 5 and are organised into four thematic pillars that taken together define the means and processes by which electrification expansion will be implemented.

**Table 5.** Summary of Malawian national electrification strategy pillars (adapted from [2]).

| NES Pillar[a] | Strategic Elements and Mechanisms |
| --- | --- |
| **Pillar I—Institutional** | • Identify roles and responsibilities of the grid and off-grid electrification implementation agencies.<br>• Develop and implement capacity-building programs to strengthen electrification stakeholders at all levels of the value chain. |
| **Pillar II—Policy and Regulatory** | • Define the minimum level of service with which access expansion will be measured.<br>• Adopt sound licensing, quality of service standards, fiscal exemptions and material/equipment standards required to support sustainable off-grid electrification; define connection fee policy for low-income grid consumers.<br>• Scale-up mini-grid and standalone off-grid system development. |
| **Pillar III—Technical and Planning** | • Identify power supply shortfalls that may impact grid densification and expansion planning on a temporal basis with which ESCOM and EGENCO can identify power supply options.<br>• Establish a least-cost geospatial planning framework for on- and off-grid electrification.<br>• Evaluate and establish low-cost electrification design standards. |
| **Pillar IV—Financial** | • Promote affordable access to electricity service for both grid and off-grid electricity services.<br>• Develop a financing plan to support the electrification expansion goals. |

### 2.2.6. Removal of VAT and Import Duties on Solar Electrification Products and Equipment

Recent economic and revenue-related policies of the GoM have sought to address the issues of unaffordability of RES, especially those targeted at improving electricity access in poorer households and communities as identified in the NEP. To support off-grid RES implementation goals, with solar systems the predominant RES in use in the country, the GoM has exempted the payment of VAT on all solar equipment and goods. Furthermore, in its 2022/23 budget, all import duties and related tariffs (i.e., excise taxes) on solar products, such as solar lights and solar fridges, were scrapped. This is with the recognition that there is currently no sustainable production capacity for such systems; hence, the expectation that there will be a reliance on their importation in the short-to-medium term while electrification expansion goals are pursued. Although overall government revenue is expected to reduce in the short term with the removal of such VAT and import duties, a report assessing the impact of tax incentives on solar implementation in Malawi [33] showed that a twofold increase in the overall direct government revenues was forecasted compared to scenarios where duties and VAT were applied on those products. However, the recent 25% devaluation of the local currency (Malawian Kwacha, MWK), and its continued depreciation (especially against the USD), has meant that the actual prices of these RES products (especially solar) have seen significant price increases as opposed to the decreases that would be expected from such policy implementation. For example, Table 6 compares the prices (in MWK—1 USD = MWK 1036 (as at 22 February 2023)) for selected solar modules and products prior to the duty removal policy coming into effect (2020) and afterwards (2023 estimates).

These increases correspond to price increases in the range of 90–200% of the costs of similar solar products in the local market in less than 2 years. This is despite average earnings not significantly increasing in the same time period. Despite the intentions targeted by the GoM to influence a favourable market price for solar modules and products with the removal of taxes and tariffs associated with such products, and hence stimulate the uptake of such solutions to improve energy access, other economic policies (i.e., currency devaluation) have significantly limited the impact of such measures, and might have negative consequences for off-grid RES implementation in the short-to-medium term.

**Table 6.** Price comparison for solar products in 2020 vs. 2023.

| Description | 2020 Price (MWK) | 2023 Prices (MWK) | Potential Load |
|---|---|---|---|
| **Standalone Solar PV System** | | | |
| **4 × 100 W panels + 1 KVA/12 V inverter + 2 × 100 Ah batteries** | 880,000 | 2,250,000 | 10 bulbs, I router, phones (2 laptops) |
| **4 × 265 W panels + 2 KVA/24 V inverter + 2 × 100 Ah batteries** | 1,691,000 | 3,256,700 | TV Home entertainment system (or 2 computers), 20 bulbs, I router, phones |
| **Power Back-up System** | | | |
| **1 KVA/12 V inverter + 100 Ah batteries** | 487,000 | 1,198,000 | |
| **2 KVA/24 V inverter + 2 × 150 Ah batteries** | 820,000 | 2,478,500 | |

## 3. Implementation Considerations and Relevant Technologies to Facilitate Energy Access

Prior to discussions on the potential implementation and adoption of suitable mechanisms to improve electricity availability to areas which currently do not have access, an understanding of possible electricity requirements and consumption patterns is necessary to inform the best-suited solutions. Ideally, such solutions should be fit to meet the economic, operational and use capacities of the targeted users.

The multi-tiered framework approach put forward by the Energy Sector Management Assistance Program (ESMAP) and Sustainable Energy for All Program (SEforALL) [13] measured and used household electricity access as a continuum of improvement and reflects on all aspects of electricity supply. The multi-tiered framework considers different energy sources (i.e., electricity and fuels) and services (i.e., lighting, refrigeration and space cooling), electricity availability, affordability and consumption, and uses these metrics as a basis to identify energy use segments (tiers) as shown in Table 7.

**Table 7.** Multi-tier energy framework to measure access to household electricity supply (adapted from [13]).

| | ESMAP/SE4ALL Framework Tiers | | | | | |
|---|---|---|---|---|---|---|
| **Characteristic** | **Tier 0** | **Tier 1** | **Tier 2** | **Tier 3** | **Tier 4** | **Tier 5** |
| **Power capacity rating (daily watt-hour -Wh requirement.** | - | Min 12 Wh | Min 200 Wh | Min 1.0 kW | Min 3.4 kW | Min 8.2 kW |
| **Supported appliances and energy usage** | - | Lighting, phone charging, light entertainment, i.e., radio | General lighting, phone charging, television/fan (if needed) | Tier 2 + medium-power appliances (i.e., refrigeration and mechanical loads) | Tier 3 + high-powered appliances (space cooling and cooking) | Tier 4 + very-high-powered appliances |
| **Typical electricity supply technologies** | - | Solar lanterns, solar chargers | Small solar home systems (SHS), rechargeable batteries | Medium SHS, fossil fuel generators, mini-grid | Large SHS, central grid, fossil fuel generators | Central grid, large SHS, fossil fuel generators |

With only 14% of the population currently having access to electricity, this means that 86% fall under Tier 0. Proposals to improve electricity access to communities not covered by the national grid or supplies from decentralised mini-grids or household electricity generation schemes would therefore ideally target providing tier 1–2 capacity

for households (at least in the short and medium term). This will be with the intention of providing entry electricity access for such households while meeting their basic needs for space lighting and communication.

To meet this goal, appropriate, adequately sized, economical RE technologies are required. Here, renewably driven electricity generation systems in the pico- or microscales can be applied to meet the electricity access goals. In the context of decentralised installations in Malawi, the main renewable resources considered to meet RE generation both for household and community levels are solar, hydro (due to a predominance of water bodies), wind and biomass. The specific RE technologies which will be considered are highlighted below (Table 8).

**Table 8.** Potential renewable energy technologies to be applied to afford electricity access.

| | Use Scale | |
|---|---|---|
| **RE Resource** | **Household** | **Community** |
| **Solar** | Pico solar PV systems (usually in 0.1–30 W range) Micro solar PV home system | |
| **Wind** | Micro-wind generator system (i.e., 200 W–1 kW systems) | |
| **Hydro** | | Pico hydro-power systems (<5 kW) Micro hydro-power systems (5–100 kW) |
| **Biomass** | Biodiesel or ethanol generators | Biodiesel generators Biogas plant with generator |

These systems are often available commercially from a wide range of suppliers. The pico-solar systems are normally small independent devices (usually plug and play) mainly used for lighting and, where modular configurations are available, can also be used to power smaller devices or charge a battery. What is designated a pico-solar system varies and is usually dependent on industry and manufacturer naming convention and marketing, as well as the local use environment.

Pico-solar systems are currently the largest and most widely used technology to attain energy access penetration in Malawi and the Sub-Saharan African region, especially on a household level. Potential electricity access proposals and mechanisms will most likely include the use of such systems, owing to their affordability, ease of use and comparative availability compared to other systems.

## 4. Ongoing Electrification Initiatives in Malawi

### 4.1. Malawi Rural Electrification Programme, MAREP

As a means to accelerate rural electrification, in 1980 the government established the Malawi Rural Electrification Program (MAREP) to increase electrification access for people in peri-urban and rural areas from 18% to 80% by the target year 2035 using the global tracking framework, GTF [29]. The program is managed by the MoE and ESCOM. The programme is funded through a 4.5% levy on all energy sales (i.e., liquid fuels, ethanol, liquid petroleum gas [LPG] and electricity) [13].

The main mechanism implemented by the programme was to facilitate the expansion of the national grid to communities that had been previously not covered, and an increase in the power generation capacity to meet this objective. Since its inception, eight phases of the program have been implemented [29]. This involved extending power distribution lines to district administration centres, major trading centres, tobacco-growing areas and the development of the 4.5 MW Wovwe Hydroelectric Power Plant. Under MAREP, 836 district administration and trading centres in rural areas in Malawi were connected to the grid [25]. However, despite the progress made since the launch of this initiative, access to the electricity grid remains quite low nationally, with only 3.9% of rural households

connected to the national grid. This implies that the main objective of the program is far from being fully met.

As a way of addressing this challenge, the Malawi government, through the Department of Energy Affairs in collaboration with ESCOM, revised the implementation of MAREP via the implementation of the NDAWALA project, which is focused on, and aims to specifically increase the connectivity of rural households. The NDAWALA project was introduced to assist low-income households with making grid connection possible through the provision of wiring services and connections of such households, which was reported to be an issue for households not connecting to the grid even when access was available. This is carried out through a soft loan of the equivalent of USD 67.57. The loan is deducted over an agreed period when customers are buying energy units, and 40% is deducted from every purchase made [34]. Additionally, consistent with the NEP, MAREP has been focused on the use of decentralised renewable energy systems, i.e., mini-grids and solar home systems (SHS), in areas where national electricity grid extension is deemed to be economically not viable.

### 4.2. Electrification Initiatives by Private Sector

As already alluded to, the MoE formulated enabling policies to bolster private sector involvement in electrification programs across the country [29]; for instance, through the Increasing Access to Clean and Affordable Decentralised Energy Services (IACADES) program, with funding from the UNDP, Global Environmental Facility (GEF) and the Scottish Government. The support has been used for grid extension, implementation of a 100 kW power plant and capacity building. With the support from MoE and the Scottish Government, the Mulanje Electricity Generation Agency was instituted and has managed to power 402 households, 3 schools and 24 teacher houses [2]. This initiative will be assessed more in the next section (Section 5.5). Through the same support structures, the MoE also collaborated with Community Energy Malawi, a privately run NGO, to install a solar-powered mini-grid in the central region part of Malawi, Mchinji, Sitolo village. The solar village has an installed capacity of 80 kW and is currently supplying electricity to 149 households and businesses [35]. The mini-grid is operated by Community Energy Malawi.

Further to this, in 2016, MoE undertook electricity sector reforms which created independent power producers and privately-owned companies which generate electricity and feed it to the national grid. This initiative has resulted in the installation of the 60 MW grid-connected Salima solar power plant. Other grid-connected solar power plants are under construction in the central region districts of Nkhotakota. Once completed, these solar PV power plants will add 86 MW to the national grid [23]. Additionally, off-grid renewable energy systems, mostly solar-powered systems, are offered by commercial companies registered with Malawi Energy Regulatory Authority (MERA). An estimated 7000 off-grid PV systems were installed in 2012, but little is known about the systems which are in operation [36].

### 4.3. Malawi Electricity Access Project (MEAP)

The GoM, with financial support from the World Bank, is currently implementing the Malawi Electricity Access Project (MEAP) to improve electricity access to households. The proposed project aims to connect 280,000 households, small and medium enterprises, schools, administrative buildings, and health facilities within proximity to the existing grid network upon its completion in the next 2 years [19]. This will increase the electrification rate from the current 11 to 20% by the project's completion in 2024. Additionally, part of the funds will be used to connect less privileged and poorer households in rural and peri-urban areas across the country.

## 5. Decentralised Renewable Energy Systems (DRES) Case Studies in Malawi: Progress and Challenges

The successful installation and operation of decentralised off-grid systems, especially those driven by renewable resources (i.e., RES), is an important factor in the potential electrification of the rural areas of Malawi. Several projects are implementing such off-grid technologies driven by the Malawian government and NGOs; for example, the Sustainable Energy for all (SE4All) campaign in Malawi with the initiative village power programme, or the UNDP and the MoE's sustainable energy management support project. These projects are run co-jointly by different organisations. There are also projects run by private investors and companies. This includes the 1 MW hydropower micro-grid managed and operated by the Lujeri Tea Estate, which they use to power the industry's energy requirement, with the excess power exported to the surrounding local communities [35]. Little information is commonly available regarding the efficiencies, economics and maintenance of this and similar projects. Table 9 lists some of the projects of previous and present mini-grid initiatives as reported in [35].

**Table 9.** Summary of past and present mini-grid initiatives in Malawi (adapted from [35]).

| Name and Location of Mini-Grid | Key Stakeholders and Funders | System Description | End Users and Business Model | Status | Notes on Successful or Challenging Aspects |
|---|---|---|---|---|---|
| **MEGA, Mulanje** | MMCT [1], Practical Action SG, Sgurr [2] | Hydro 80 kW | Domestic | Active since 2014 | Only breaks even after 5 sites are installed, heavily reliant on funding. |
| **SE4RC: Nyamvuwu, Chimombo in Nsanje district (30 kW and 15 kW, respectively) and Mwalija and Oleole in Chikwawa (55 and 30 kW)** | PAC [3], CARD [4], FISD [5] | 55, 30, and 10 kW | Domestic, Irrigation | Active since 2018 | Improved access to modern energy services has contributed to better well-being. Enhanced community participation and skill transfer. Increased business operation hours and study time in the evening. Crop production has increased through irrigation schemes. |
| **Sitolo, Mchinji** | CEM [6], CES [7] | Solar 80 kW | Domestic | Active since September 2020 | Financed by UNDP, community participation, skills transfer commercialization and entrepreneurship development strategy. |
| **Solar Village Mini-grids, Nkhata Bay, Nkhotakota; Chiladzulu; Mzimba; Thyolo, Ntcheu** | GoM | Hybrid (solar and wind) 35 kW in all sites | Domestic | None working currently since 2012 | No community participation during implementation. Lack of financial and business model. No skills transferred to communities. Lack of PUE activities. |

**Table 9.** *Cont.*

| Name and Location of Mini-Grid | Key Stakeholders and Funders | System Description | End Users and Business Model | Status | Notes on Successful or Challenging Aspects |
|---|---|---|---|---|---|
| **Likoma Island** | GoM | Solar–diesel generator hybrid system | Domestic and institutional | Still active with 24 h supply daily | |
| **Usingini** | PAC | Hydro (300 kW) | Domestic and commercial | The project was abandoned since the site was targeted for grid electrification by MAREP | Financed by UNDP, community participation, skills transfer commercialization and entrepreneurship development strategy. |
| **Mthengowathenga** | Roman Catholic Church | Solar Mini-grid (50 kW) | Domestic and commercial | Active since 2017 | Appreciable reduction of energy costs. Reliable and sustainable energy. In the hospital, which is connected to the public grid, longer power cuts had been observed daily with system use, however this has been minimised recently. |
| **ST Gabriel** | Roman Catholic | Solar–diesel Mini-grid (35 kW) | Domestic and commercial | Active since 2017 | The costs for public electricity and fuel for the two diesel generators are a significant financial burden. Reliable 24 h energy supply. Programmable, fully automatically working system, switching on and off, according to energy demand. |
| **Nkhata Bay Hospital** | GoM | Solar Mini-grid and solar geyser | Institutional | Active since 2015 | Programmable system automatically guarantees a 100% safe and uninterrupted energy supply with high ecological sustainability and economical use of the available energy sources. |
| **Dedza Microgrid** | United Purpose, University of Strathclyde | Solar Micro-grid (5 kW) | Domestic and Productive Users | Feasibility study complete | A successful business model relies on CAPEX funding; however, smaller capacity means lower upfront costs. |

[1] MMCT: Mulanje Mountain Conservation Trust; [2] Sgurr is a Scottish (Glasgow-based) energy company; [3] PAC: Practical Action; [4] CARD: Churches Action in Relief and Development; [5] FISD: Foundation for Irrigation and Sustainable Development; [6] CEM: Community Energy Malawi; [7] CES: Community Energy Scotland.

The following subsection sections looks specifically at some previous and current DRES projects implemented in Malawi, and uses them as a case study to provide an analysis on the causes of failure or underperformance, as well as the successes, of such projects. It is essential to learn from past projects in order to develop a successful systematic framework for evaluating current and future decentralised RE projects.

*5.1. The Solar Village Concept*

This initiative involved the installation of a micro solar–wind hybrid system set up to generate electricity in large enough quantities to supply between 100 and 150 households, a school and a trading centre. The communities and villages receiving these systems were then referred to as "solar villages". The solar–wind hybrid system had an installed capacity of 25 kW and available capacity of 20 kW [22].

In 2007, using this mechanism, a total of six solar villages were commissioned on the following sites: Kadzuwa in Thyolo District, Chigunda in Nkhotakota, Elunyeni in Mzimba, and Mdyaka in Nkhatabay, Kadambwe in Ntcheu and Chitawo Solar in Chiradzulu District [26]. The installed capacity for the solar–wind hybrid system each cost USD 60,000.00 (2008 prices), with a total of USD 360,000.00 used for the system development and execution in the six solar villages. In terms of financing and payment, some villages were supplied free of charge while others paid USD 0.20–0.50 (per household connected) every month to a local fund, used for paying an operator of the plant, the security guard (to prevent theft of the infrastructure) and some minor repairs. An additional consumer cost was USD 2 per connection including wiring materials or USD 1 in cases where wiring accessories were not included.

The concept of the solar village involved the government hiring a contractor to construct and install the renewable electricity generation facility. Upon completion, the facility is handed over to the government. The government enters a one-year contract with the contractor for the repair and maintenance of the facility. The government then hands over the facility to the concerned community. The community assumes ownership and is expected to manage the facility. The community is then expected to establish a committee to be responsible for the day-to-day management of the facility, revenue collection and repair and maintenance beyond the one-year contract. The labour and employment opportunities related to the plant operation, safety and maintenance are also expected to be drawn from the local community. For example, one member of the community is employed as the operator of the facility while another is employed as the security guard.

5.1.1. Successes

The findings made by [26] indicate that solar villages registered the following successes: a high sense of ownership of the facilities such that people displayed their acceptance of the facilities that had been established by forming committees to manage them including their daily operation, revenue collection and repair and maintenance. Additionally, the community members were willing to pay for the electricity despite some cases of defaulters.

5.1.2. Challenges

Studies by [35] indicate that it was costly to install the solar villages and the operating costs were too high, such that the locals could not afford to repair and maintain the system, especially the batteries. Furthermore, there was a lack of dedicated funding for sustaining the programme, such that currently, no solar village is operating. The other challenge was the unsustainable electricity pricing mechanism, such that electricity tariffs were equivalent to monthly expenditures on torches, candles and other lighting sources used by the community members.

*5.2. Decentralised Systems Strategy: The Barefoot Engineers Concept*

In 2007, the Barefoot Engineers concept was introduced in Malawi through a project that was financed by the Indian Government in collaboration with the Malawi Government

through the Ministry of Foreign Affairs and International Cooperation [26]. The objective of the project was to disseminate solar home systems to vulnerable people in rural areas. Under this project, a total of 316 solar home systems were installed in 316 households in 4 villages in all 3 regions of Malawi. The installed system capacities were based on the number of electrical appliances a household had or intended to have, as well as the operating hours of such appliances. However, most SHSs installed were 1 kW systems. The Indian Government provided full scholarships for the training of selected rural women in solar PV engineering. These women were trained for a period of 6 months at Barefoot College, India. The last cohort of women to be trained under this project did so in 2013. For its part, Barefoot College has partners in countries where the project is being implemented. In Malawi, the partner is the Centre for Community Organisation and Development (CCODE).

### 5.2.1. Successes

Since the Barefoot Engineers project focused on the installation of solar home systems which are simple and less complex, it was easy to transfer skills to the local community and less costly, thereby making it easier to repair and maintain the systems. Additionally, there was a strong sense of ownership and community participation since each household had its solar home system. Thus, the decentralised system strategy provides a self-regulating environment as far as ownership, operation, care, repair and maintenance are concerned. However, little is known on whether all of the solar home systems installed are still in operation.

### 5.2.2. Challenges

The strategy was weakened by the fact that the first batch of the systems were given out to the selected participants free of charge. There was no provision for people purchasing the systems on their own. This means that the rest of the villagers looked forward to "receiving" rather than "buying" the systems. A knock-on community uptake of the concept and the solar home systems was therefore observed not to have occurred. Furthermore, due to its dependence on donations, the project had very limited coverage. For example, Chitala Centre comprises 19 villages but only 4 were covered by the project [22].

### *5.3. The Sunny Money Concept*

The Sunny Money concept, operated by the UK-based charity Solar Aid, works as a social enterprise which identified the need and potential market for affordable lighting systems in rural areas, and aimed to provide a sustainable market model to meet that need. With the concept, Solar Aid imported solar PICO lights and sold them to customers [26]. The business model applied saw the use of an innovative marketing system to deliver new products to hitherto unknown markets. With this scheme, Sunny Money involved the participation of local individual dealers and entrepreneurs across the country to increase the reach of these products. This is prompted by investment in the collaborative marketing of solar PV technologies to end users.

### Lessons from the Concept

The significance of the Sunny Money concept is that it contributes to the development of the market infrastructure for the transfer of solar PV technologies. To be specific, the model highlights "dealerships" as a marketing strategy that enables the transfer of solar PV technologies deep into rural areas [26]. The dealership which builds local capacity for the supply of relevant products is therefore an important feature that should become one of the pillars of a sustainable strategy for the transfer of solar PV technologies to the rural areas

### *5.4. The Empower Concept*

Empower Inc. is an Australian NGO that operates the Empower Malawi concept (a social enterprise that seeks to achieve energy access improvement), as well as the

promotion of a savings culture. Empower Malawi facilitated the distribution of solar lanterns for lighting in the remote community of Kapita in Mzimba District in 2010. The project was first piloted in Zatuba Village in the South-East Mzimba District, Northern Region of Malawi [35]. The project was designed in such a way as to empower the people economically, first and foremost.

In 2017, Empower Malawi partnered with the NextEnergy Foundation to establish five "independent" energy hubs with the aim of providing electricity to hard-to-reach rural communities in Northern Malawi. The hubs were positioned to bridge the finance gap for very-low-income community consumers through a service offering pay-per-use access to energy goods and services. These included the charging of mobile phones, batteries and other portable devices, periodic rentals of solar lights and other solar products and carrying out minor repairs. These hubs therefore allowed consumers to access such services without incurring the relatively significant costs associated with their outright purchase.

The energy hubs further worked in conjunction with local schools across Northern Malawi. The schools were used as hub bases, while they in turn were provided the benefit of free lighting (which allowed remedial classes to be run late in the evenings, thus improving the services provided by the schools). With regard to the empowerment goals through encouraging a savings culture, Empower further provided seed money of USD 10,000.00 to facilitate its objectives. One of the first activities was to establish a village bank to engender self-reliance in the area through the promotion of a saving culture in the community. According to [26], in addition to promoting saving, the bank aimed to provide business and development loans in a region that has no financial services within a 50 km radius. At the community level, savers earn 7 percent interest on their savings while borrowers pay 20 percent interest on the loans. Empower engaged a local training and consultancy provider, namely Business Expansion and Entrepreneurship Development (BEED), to train members of the community in business and entrepreneurship, as well as to establish a Village Revolving Fund. The fund would be the capital base for the provision of loans. One outstanding feature of the project design is the absence of donations.

### 5.4.1. Successes

The Empower concept is very close to being a sustainable strategy for solar PV technology transfer. The potential for sustainability lies in the fact that there are no donations involved. This means that any capital injections by Empower Inc., referred to as seed money, will be repaid. The other favourable aspect of the model is that it has a built-in economic empowerment component, which enables participating communities to develop the financial capacity to purchase solar lanterns, among their other needs. The model highlighted "economic empowerment" as an important pillar of a sustainable strategy for the transfer of solar PV technologies in remote areas of Malawi.

### 5.4.2. Challenges

There is no effective supply chain for the technologies built into the project design. Communities cannot usually support the establishment of the hubs on their own, and individual consumers were usually unable to afford the solar lanterns, relying on the sponsors to keep supplying them. This is where the sustainability of the project is compromised.

### 5.5. Mulanje Electricity Generation Agency

Mulanje Electricity Generation Agency, MEGA, is located in Mulanje district, Malawi, and is considered the first mini-grid in Malawi. The concept of a community-based mini-hydropower project was introduced in 2008 and led to the establishment of the Mulanje Energy Generation Agency (MEGA) in 2011 by three founding partners [35].

The installed system capacity is 80 kW. The MEGA project is managed by Powering Development in Mulanje (PDM) with financial support from the European Union and the Organisation of the Petroleum Exporting Countries (OPEC) Fund for International Development (OFID), the Scottish Government, Mulanje Renewable Energy Agency, MUREA,

Practical Action, UNDP/GEF and other funders. Being a socially oriented company, MEGA does not seek to maximise profits. The MEGA business model aims to achieve economies of scale for central operations by developing multiple sites [37,38].

### 5.5.1. Successes

The hydro scheme mini-grid supplies power to 740 households, 5 primary schools, 24 teacher houses and 32 businesses. Other entrepreneurs intend to venture into businesses such as maize mills, welding and machine shops, carpentry, bakeries and hair salons. The mini-grid has also improved the services at the local health centre, as the clinic is now fitted with lights and a has a vaccine refrigerator and sterilizer. Prior to electricity supply, expectant mothers brought candles to the clinic in case deliveries occurred at night. Based on the project successes, the PDM intends to install a 100 kW hydro power facility to further afford access to 400 additional homes using the mini-grid mechanism.

### 5.5.2. Challenges

MEGA faces specific difficulties because the electricity tariffs are constrained by consumer ability to pay. Another difficulty is the lack of enough human and technical resources in the area.

## 6. Obstacles and Issues Affecting DRES Implementation in Malawi

Even though improving energy access (especially to poorer rural communities) appears to be a high-priority policy issue in Malawi, with its potential strategies and actions put forward in different government policy documents, the actual realisation or the mechanisms to achieve them have not been fully implemented or effected to ensure adherence. There also appears to be a current lack of resources to afford the realisation of such electrification goals. On a governmental level, it can be observed that even with the institution of the rural electrification fund which was positioned to assist the co-funding of DRES in remote rural communities, the inadequacy of this fund to successfully finance the national targets has been highlighted. On a community level, potential DRES projects also suffer owing to the earning power of inhabitants, especially since initial investments for such projects are normally too large to be covered wholly by such communities. A survey by the Irish-Research-Council-funded CEANGAL project in Malawi [39] found that over 55% of the interviewed respondent households earned less than MWK 50,000 monthly (i.e., less than USD 48). More flexible funding mechanisms and financing resources relevant to the particular local scenarios are therefore needed.

While there has been some noticeable impact in improving electrification using decentralised off-grid systems through the implemented projects, the Malawian government's operational systems do not currently appear to promote the setup of these systems. For example, in cases where a micro-grid is targeted to be established, policies associated with obtaining a license for the installation of a microgrid can be problematic. This is especially the case with the long waiting times (with reports of sometimes up to 2 years) generally encountered to obtain permissions. The price of the registration and license fees (especially for 1–5 MW RES systems generation) is also observed to be significant and potentially a hinderance, especially when compared to other SSA countries. The registration and license fees are USD 145 (for private consumers) and USD 483 (for commercial purposes, respectively (2023 estimates), which contrasts the situation in Senegal and Burkina Faso, where there is no cost for similar licenses [29,35]. In Kenya and Tanzania, there is no need for a license for microgrids producing less than 1MW, regardless of their purpose. This may be one of the reasons why the implementation of these systems is more developed in these countries.

Another major issue limiting an increased uptake of DRES solutions in Malawi is the eventual costs or use fees related to the power generated by such systems. This is exemplified by the tariffs currently proposed by the company ESCOM. The common price of electricity was USD 0.09/kW or in the package of USD 33 for 365 kW/h in 2020. This

price is set for ESCOM-distributed electricity, regardless of the source. Tariffs set by the company have seen increases in the last decade. The main concern with such increases is that over 60% of the households in Malawi have been identified as being unable to afford these prices, which are higher by 20% than the average household earning [13]. To exacerbate this issue, the World Bank, in its report assessing the electricity pricing situation in Malawi, estimated that the electricity prices were not "reflective" of the real price of production. They calculated that the "ideal" tariff would be USD 0.13/kW, which would be more reflective of the production costs (*note: this refers to 2016 values, with expectations that this estimate will be higher in 2023 due to inflation*). In such scenarios, where the ideal tariffs are effected, most households will struggle to pay their electricity bill, leading to further apathy regarding the need to be connected to an electricity source and the accruable benefits [13]. To help local communities to pay for electricity consumption from decentralised mini-grids, some plans have been developed offering power using a Pico solar system (3 lights, and charging a phone) for one year at USD 42 and with SHS (four lights, charging phone and radio) at USD 102/year [13]. Other systems, such as Pay as You Go, propose to pay by phone-in instalments.

In relation to the certification of infrastructures available to meet the actualisation of implementing RES systems, there is a lack of mechanisms to verify genuine and reliable systems. A recent survey established that a vast majority of the solar panels (Pico and larger solar home systems) found in Malawi were copycat/counterfeit products, with their operational lifetimes being significantly shorter than the genuine systems [40]. The "fake" solar panels and modules usually came with no warranties, and if they were damaged, no repairs or replacement was possible. There was a lack of education for the population by the government or certification authorities in Malawi to distinguish a fake from a certified panel. Even though MAREP under its decentralised RES drive requires that solar panels have at least a warranty, most retailers in Malawi do not offer it. A significant number of such modules arrive in the country "pre-used" from South Africa. Interested off-grid RES operators and consumers in Malawi are therefore vulnerable to a lack of regulation and protection [40]. In addition, the poor-quality materials in such systems could result in a lower yield of solar panels and reduced confidence in decentralised RES solutions.

Surveys conducted by McCauley et al. [14] suggested that investors and stakeholders were not willing to commit substantial investments that would have supported the proliferation of DRES and off-grid systems, due to the non-availability of strong robust policies implemented by the government. There is a lack of knowledge of the current situation and misconceptions about the country. Investors are wary of the political and economic situation of the country, seeing it as being risky and not stable enough for guaranteeing payback or even a successful sustained operation of such projects in cases of a charitable donation to meet electrification goals.

Furthermore, there seemed to be an over-reliance on the expectations of the benefits of energy access through the national grid, even where such access was not currently available, and where there were no plans to achieve this in the short-to-medium term. Some respondents interviewed thought that using mini-grids could hinder the development of the national grid and identified that such DRES solutions were "unsustainable". Some investors even identified a preference to develop the supply of energy coming from a neighbouring country such as Mozambique [14]. There was also a fear of the inefficient management of energy by ESCOM, and this was used as a reason for the lack of confidence in further investing in the decentralised RES sector. Nevertheless, although the distribution and the generation of power have been separated into two entities, the overall system was considered poorly managed and unreliable. ESCOM is perceived as one of the riskiest companies in Africa. Indeed, they suffer from a lack of support from the government and their reputations precludes them from securing extensive funding [14].

There is currently a lack of dissemination and knowledge transfer activities specifically aimed at rural communities with low electricity access opportunities, and currently at a distance from the national grid, as identified in [13]. A study carried out in the south of

Malawi in the city of Blantyre revealed that 32% of households were powered by solar energy. The majority of the respondents had a higher education with a secondary degree (52%, compared to 8% with a primary degree) [37]. It emphasised that more communication should be effectuated by the local government regarding the advantage of this technology and reduce the misconceptions about solar energy and RES in general, which might in turn stimulate collective or individual action based on the knowledge of the benefits.

One of the major issues in Malawi was the significant lack of human resources and the local capacity to support the realisation of the decentralised RES actualisation goals, including the support structures to afford the installation, operation and maintenance of the potential off-grid mini-grids and standalone RES systems. For example, it was observed that after the installation of 5000 solar home systems in 2000, at least 50% of them were not working 3 years later [39]. The repairs of those systems were greatly impaired with a lack of trained local skillsets to provide repairs to this equipment; hence, they ended up being discarded. Some skills to operate DRES are complex and highly qualified persons are required to maintain some systems. Although there has been increased participation of private companies, NGOs and the government sector in improving decentralised RES generation capacity with the installation of infrastructures, there has been a lack of intensive support to facilitate the continuous training of local personnel to operate and maintain these systems [41].

## 7. Recommendations and Commentary

The perceived mismanagement of Malawi institutions and uncertainty around changing government policies could have a negative effect on the development of mini-grid and DRES projects by discouraging potential interest in the sector, not just of international and local investors, but also in community members who might have wanted to undertake such schemes. The government should work closely and in collaboration with the private sector investors, NGOs and community stakeholders to develop a more efficient, reliable and workable practical strategy; such a strategy should be easily understood and interpreted by all stakeholders, and the necessary requirements and guidelines made clear to ensure transparency.

More focused education targeting local communities is also required to address the continued preference for a connection to the national grid for electricity access as opposed to the use of DRES systems. The lack of strong examples of sustained electricity access using decentralised systems is mainly to blame for this. The use of international and national case studies which have been successful in guaranteeing continuous electricity access should be shared. The benefits of RES systems and their potential impact on the livelihoods and economies of such communities should be highlighted as well.

Malawi's government could, through a collaborative process put forward a framework to support a fairer, more sustainable and considerate tariff structure, that will not only encourage future investments, but will also facilitate an improvement in the willingness to participate and pay tariffs related to such schemes. The government could follow the example of the ReFIT schemes successfully applied in Tanzania where a standardised power purchase agreement is used [26]. This decision could enhance the deployment of decentralised RES [15,42].

Although the NEP clearly indicates an intention to support local capacities to distribute and supply RES, especially solar modules, to assist electricity access goals, there is currently a significant lack of suppliers and distributors nationally for these products, with the few available mainly concentrated in the urban areas and with no regional offices or sales centres available. The difficulty in purchasing and finding solar panels in the Dedza district of Malawi was demonstrated in [37], with only 13% of the retailers selling related products having such modules available. Furthermore, in cases where these were stacked, there is no information on whether these solar panels were certified as mentioned earlier [43]. Therefore, the government, private companies and NGOs should help with mechanisms to ensure the availability of RES modules, including solar systems for purchase. This could

be through the provision of credit facilities to allow the wholesalers and distributors to effectively purchase and have such products available. The government could also play a more significant role in supporting product availability by bulk-buying the modules and selling through registered outlets in closer proximities to the rural communities, and at cost price.

To address the observed lack of knowledge and working skill sets and expertise on the concepts, operations and maintenance of RE generation systems which negatively impacts the operation sustainability, more concerted efforts by all stakeholders are needed to facilitate the training and knowledge transfer to support the RES implementation goals. More specialised training and workforces to equip individuals with required skills are needed. Furthermore, targeted community support centres providing specialised services including information on ideal RES solutions, to provide advice on system operation and to carry out infrastructure repairs are needed. This could be through the establishment of functional local service hubs for local stakeholder engagement, which would ensure the continuity of established RES projects and use, act as a regional information point for future communities' interests in RES implementation and provide training and support to drive the continued process of RES uptake.

## 8. Conclusions

SDG7 and Se4All programs financed by the United Nations face the main challenge of improving the rate of electrification in low-income countries, emphasising their actions in the areas with difficulties in accessing electricity. In the SSA region, Malawi has the lowest rate of electrification, with approximately 14% of the total population having any access to an electricity source. The national grid is the main source of access, with hydropower being the primary source of electricity for the national grid. Despite the low penetration of the grid, the country cannot currently cope with the electricity demand, leading to daily long blackouts usually reported nationwide. Such power outages are more common in dry seasons (with lower water levels in reservoirs). To meet the objective of supplying at least 80% of the population by 2030, the generation of electricity using off-grid using mini-grid or decentralised renewable energy systems have been increasingly pursued. Solar photovoltaic is currently the most widespread applied RE technology in Malawi.

While more government action is required to facilitate a more favourable market to encourage an improved uptake and use of RES especially in rural areas, the availability of more practical funding mechanisms to support such goals are required. With Malawi being one of the poorest countries in the world, fairer tariff rates are needed in order for most of the underserved population to benefit from the intended electrification actions using DRES. Furthermore, government, community stakeholder and private companies should collaborate for the education of local communities on the advantages of setting up DRES to ensure that the benefits accruable with the use of such systems will be best understood by all the stakeholders.

**Author Contributions:** Conceptualisation E.A.E.; Materials organisation: P.Y.S. and E.A.E.; Text preparation and writing—original draft preparation, P.Y.S., E.A.E. and T.R.; writing—review and editing, E.A.E., P.Y.S. and T.R.; supervision, E.A.E. and G.T.G. All authors have read and agreed to the published version of the manuscript.

**Funding:** The research leading to the publication received funding from the Irish Research Council COALESCE (Collaborative Alliances for Societal Challenges) Funding Strfand 2B—Better World Awards 2020 in partnership with the Department of Foreign Affairs, for the CEANGAL (Community Based Decentralised Renewable Energy Systems and Supporting Structures for Improving Electricity Access in Low-income Countries) project, with project funding number COALESCE/2021/41. The CEANGAL project is a collaborative policy project between Atlantic Technological University Sligo, Ireland (ATU Sligo) and Malawi University of Business and Applied Sciences (MUBAS), which looks to put forward a practical framework for improving decentralised renewable energy implementation in Malawi. The views and opinions expressed in this article do not necessarily reflect those of the Irish Research Council and the Irish Department of Foreign Affairs.

**Conflicts of Interest:** The authors declare no conflict of interest. The funders had no role in the design of the paper; in the collection, analyses, or interpretation of data; in the writing of the manuscript, or in the decision to publish the results.

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
