# Peer review of "Improving Energy Access in Low-Income Sub-Saharan African Countries: A Case Study of Malawi"

_energies, doi:10.3390/en16073106_

Round 1
Reviewer 1 Report
Authors presented a nice review & commentary study focusing on improving & evaluating Malawi's energy access. Besides the clear motivation, the recent qualitative and quantitative data about the country are also useful, particularly after seeing the impacts of both COVID and war conflicts. The work has a broad concept but provides a smooth reading a a policy study, but there are some minor points for a better organization. Please see them below:
1) Please remove the parenthesis "especially electricity" from the Abstract.
2) Paragraphs 1 and 2; 3 and 4; and 5-6-7-8-9 can be combined. The last paragraph can better summarize the main perspective of the review study after the sentence " this paper intends...".
3) Figure 1 should be improved from the font sizes in the figure. Authors can also zoom-in with a sub-figure to better point our the important regions.
4) Can authors add the interest and inflation rates of the corresponding years of commissioning in Table 1? It could give a useful insight to the readers. World Bank data can be used. For the plants that were commissioned in a long time period (e.g. the first three rows), capital recovery factors can be used.
5) After Table 1, the following 4 paragraphs can be combined. Then, the following paragraphs can also be combined until Table 2.
6) The font style should be in Palatino Linotype in Table 2 (MDPI requirement).
7) After Table 3, two paragraphs until 2.2 should be combined.
8) Until Figure 2, all paragraphs should be combined.
9) From 2.1.1. to Section 3, all paragraphs in each sub-sub section (e.g. 2.1.1, 2.1.2, 2.1.3, etc.) can be a single paragraph.
10) Like 3.3., 3.1 and 3.2 can also have single paragraph by combining short paragraphs.
11) From Section 4 to Table 4, all paragraphs should be combined.
12) Section 4.4., the last two paragraphs should be combined.
13) Section 4.5.1, two paragraphs should be combined.
14) Some of the references seem inconvenient for MDPI guideline, please double check them.
Reviewer 2 Report
The article is very well structured and easy to read. It has a clear objective, is well contextualized and the analysis performed is adequate and well presented. It is not a more standard article of introduction, methodology, results and discussion, but this does not mean that it is of lower quality. On the contrary, it is of great interest to read analyses in this style and they are very necessary, especially when they are coherent, as in this case. I think the article has only a few minor writing flaws, of which I have pointed out a few below (although it is likely that I have missed a few more, so a thorough review is recommended):
Line 86, 110: comma
117: double space
524 (title)
526 (full stop)
825 (space before full stop)
Reviewer 3 Report
I read with interest the article "Improving Energy Access in Low-Income Sub-Saharan African Countries: A Case Study of Malawi" and I have the following comments to make:
1) The article provides a very good general presentation of the living conditions and access to energy of the inhabitants of Malawi. Also presented are projects for the development of production capacities and electricity supply for different communities/regions.
What is missing, apart from this very good documentation of the current state and energy problems in Malawi, are the scientific aspects.
2) Then, the numbering of the tables in the article seems taken from another document, because the text talks about (for example) Table 2.4 and the actual table is called Table 4. The same for Tables 1, 2, 3, 5, 6. Regarding Table 7, it is presented in the text as Table 4.0.
3) The same observation for figures 1 and 2, referred to in the text of the article as 2.1 and 2.2.
Round 2
Reviewer 3 Report
The authors of the article "Improving Energy Access in Low-Income Sub-Saharan African Countries: A Case Study of Malawi" made the additions and corrections requested by the reviewers for the previous form of the article.
The quality of the article was thus improved.